# Clinical and Cytokine Profile in Patients with Early and Late Onset Meniere Disease

**DOI:** 10.3390/jcm10184052

**Published:** 2021-09-07

**Authors:** Maria-Del-Carmen Moleon, Estrella Martinez-Gomez, Marisa Flook, Andreina Peralta-Leal, Juan Antonio Gallego, Hortensia Sanchez-Gomez, Maria Alharilla Montilla-Ibañez, Emilio Dominguez-Durán, Andres Soto-Varela, Ismael Aran, Lidia Frejo, Jose A. Lopez-Escamez

**Affiliations:** 1Otology and Neurotology Group CTS495, Department of Genomic Medicine, GENYO—Centre for Genomics and Oncological Research—Pfizer/University of Granada/Junta de Andalucía, PTS, 18016 Granada, Spain; mcarmenmoleon@gmail.com (M.-D.-C.M.); estrella.martinez@genyo.es (E.M.-G.); marisa.flook@genyo.es (M.F.); andreina.peralta@genyo.es (A.P.-L.); lidia.frejo@genyo.es (L.F.); 2Department of Otolaryngology, Instituto de Investigación Biosanitaria ibs.Granada, Hospital Universitario Virgen de las Nieves, Universidad de Granada, 18014 Granada, Spain; 3Sensorineural Pathology Programme, Centro de Investigación Biomédica en Red en Enfermedades Raras, CIBERER, 28029 Madrid, Spain; 4Department of Neurology, Hospital Universitario Virgen de las Nieves, Universidad de Granada, 18014 Granada, Spain; jagaza16@correo.ugr.es; 5Otoneurology Unit, Department of Otorhinolaryngology, University Hospital of Salamanca, IBSAL, 37007 Salamanca, Spain; hortensiasanchez1@hotmail.com; 6Department of Otorhinolaryngology, Complejo Hospitalario de Jaén, 23007 Jaen, Spain; alharillamontilla@gmail.com; 7Department of Otolaryngology, Hospital Virgen Macarena, 41009 Sevilla, Spain; emiliodominguezorl@gmail.com; 8Division of Neurotology, Department of Otorhinolaryngology, Complexo Hospitalario Universitario, 15706 Santiago de Compostela, Spain; andres.soto@usc.es; 9Department of Otolaryngology, Complexo Hospitalario de Pontevedra, 36071 Pontevedra, Spain; ismaelaran2000@yahoo.com; 10Department of Surgery, Division of Otolaryngology, Facultad de Medicina, Universidad de Granada, 18006 Granada, Spain

**Keywords:** hearing loss, vertigo, migraine, cytokines, inflammation, vestibular disorders

## Abstract

Background: Meniere disease (MD) is an inner ear disorder associated with comorbidities such as autoimmune diseases or migraine. This study describes clinical and cytokine profiles in MD according to the age of onset of the condition. Methods: A cross-sectional study including 83 MD patients: 44 with early-onset MD (EOMD, <35 years old), and 39 with late-onset MD (LOMD, >50 years old), 64 patients with migraine and 55 controls was carried out. Clinical variables and cytokines levels of CCL3, CCL4, CCL18, CCL22, CXCL,1 and IL-1β were compared among the different groups. Results: CCL18 levels were higher in patients with migraine or MD than in controls. Elevated levels of IL-1β were observed in 11.4% EOMD and in 10.3% LOMD patients and these levels were not dependent on the age of individuals. EOMD had a longer duration of the disease (*p* = 0.004) and a higher prevalence of migraine than LOMD (*p* = 0.045). Conclusions: Patients with EOMD have a higher prevalence of migraine than LOMD, but migraine is not associated with any cytokine profile in patients with MD. The levels of CCL18, CCL3, and CXCL4 were different between patients with MD or migraine and controls.

## 1. Introduction

Patients with acute vertigo and a history of episodic vestibular symptoms represent a diagnostic challenge for clinicians [1,2]. Particularly, Meniere disease (MD), an inner ear disorder characterized by sensorineural hearing loss (SNHL), attacks of episodic vertigo, is usually associated with tinnitus [3]. Some patients with MD also have migraine headaches and they usually show an early onset of auditory and vestibular symptoms [4,5].

Although the majority of MD patients are considered sporadic [3], familial clustering has been reported in 8% and 6% of cases in the Spanish [6] and South Korean populations [7]. Several genes have been identified in familial MD by exome sequencing, the most common being *OTOG*, found in 30% of familial cases [8]. Familial cases usually show early age of onset, but some non-familial cases (sporadic) may also develop the disease in the third decade of life. The mechanisms leading to this early onset of symptoms are not well understood and they may involve genetic, epigenetic, and environmental factors.

The epidemiological association of MD with migraine and some autoimmune diseases, supports the hypothesis of a wide clinical spectrum with a cochleo-vestibular phenotype. However, the effect of autoimmunity or migraine on the onset of the condition has not been investigated. Previous studies have shown that MD can be classified according to few clinical predictors, and it is possible to define subsets within unilateral MD (UMD) and bilateral MD (BMD) patients (Table 1) [4,5].

The age of onset in patients with MD may differ according to the clinical subgroup. Interestingly, patients with MD and migraine (type 4) and individuals with MD with a comorbid autoimmune condition (type 5) have an earlier age of onset than MD type 1 or 2 [5]. Moreover, it was also observed that MD types 4 and 5 have more vertigo attacks than the other groups [5]. Thereby, individuals with BMD showed an earlier age of onset, associated with a worse score in the AAO-HNS functional scale than patients with UMD among familial cases [6].

Previous studies have reported that basal levels of proinflammatory cytokines may be increased in some patients with MD [9], but the relationship between cytokine levels with the age of onset, migraine, or autoimmunity has not been investigated. The cytokine profile may represent two subgroups of MD patients with intrinsic differences in the immune response or two functional states of the immune system in patients with MD [9,10]. The studies mentioned before had defined different clinical subgroups in patients with UMD or BMD, but patients with MD and increased basal levels of cytokines and negative autoantibodies may be considered as an autoinflammatory condition of the inner ear [11].

The aim of the study is to characterize the clinical and cytokine profile, according to the age of onset in MD.

## 2. Experimental Section

### 2.1. Patient Assessment and Selection

A cross-sectional study was designed to recruit a total of 83 patients with MD. Eighteen of them were also included in a previous study [10]. Forty-four patients were considered as early-onset MD (EOMD, age of onset < 35 years old), 39 patients were classified as late-onset MD (LOMD, age of onset > 50 years old). We also recruited 64 migraine patients and 55 healthy controls that were selected from the MeDiC database to compare clinical and cytokine profiles among the different groups. All patients with MD were diagnosed according to the diagnostic criteria for MD described by the International Committee for the Classification Vestibular Disorders of the Barany Society [3], and a clinical evaluation was performed after obtaining informed consent. Patients with migraine were diagnosed according to the diagnostic criteria for migraine of the International Headache Society [12]. Patients with diagnostic criteria for vestibular migraine [13] were excluded from this study.

The experimental protocol of this study was approved by the Institutional Review Board in all participating hospitals, and each participant signed a written informed consent before donating blood samples (PE-0356-2018). The study was carried out according to the principles of the Declaration of Helsinki revised in 2013 for investigation with humans.

A complete audiological and vestibular assessment was carried out in all MD cases, including the following variables, sex, age, age of onset, duration of the disease, type of MD (uni/bilateral), main associated comorbidities (migraine, autoimmune disorders), and cytokine levels in peripheral venous blood. Serial pure-tone audiograms were retrieved from clinical records to assess hearing loss since the initial diagnosis. Clinical features of migraine patients recruited in the study are included in Appendix A.

### 2.2. Peripheral Blood Mononuclear Cell (PBMC) Isolation and Incubation

Peripheral blood was diluted 1:1 with 1× PBS and disposed carefully into the corresponding 20:12 volume of Lymphosep, lymphocyte separation media (Biowest, Nuaillé, France). Samples were centrifuged for 20 min at 2000 rpm to separate blood content. PBMCs were collected and washed with 1× PBS twice and cultured in RPMI 1640 supplemented with 10% (*v/v*) fetal bovine serum (Biowest, Nuaillé, France) and plated at a concentration of 1 × 10^6^ cells/mL in 6-well plates. PBMCs were incubated overnight at 37 °C in 7% CO_2_. After the incubation, PBMCs were centrifuged, and supernatants were collected and stored at −80 °C until enough samples were acquired.

### 2.3. Cytokine Measurement

Frozen samples were thawed immediately prior to analysis and none of the samples underwent more than two freeze–thaw cycles prior to analysis. One cytokine (IL-1β) and five chemokines (CCL3, CCL4, CXCL1, CCL22, and CCL18) were measured using the commercially available Multiplex Bead-Based Kits (EMD Millipore, Billerica, MA, USA). The measurements were performed in accordance with the kit-specific protocols provided by Millipore, using a Luminex 200 platform (Luminex Corp., Austin, TX, USA) and read with Luminex x PONENT 3.1 software (Luminex Corp.). Two quality controls for each cytokine were run in duplicate.

### 2.4. Statistic and Data Analysis

A descriptive analysis was conducted using GraphPad Prism v8.0 (GraphPad Software, San Diego, CA, USA) for the clinical and cytokine data. Quantitative variables were displayed as the mean and standard deviation (SD). Qualitative variables were compared using Chi-square and Fisher exact test. Quantitative variables were compared using U Mann–Whitney test and the Pearson correlation coefficient. Analysis of variance (ANOVA) was used to compare the three groups of the study (MD, migraine patients, and healthy controls). The level of significance considered was *p* < 0.05. To find and remove the cytokines outlier values, we considered outlier values that were:1.5 × IQR > Q3
1.5 × IQR < Q1,
where Q1 is the median of the lower half of the data set, Q3 is the median of the upper half of the data set, and IQR is the interquartile range, the difference from Q3 to Q1, [14]. Thus, from the total cytokines’ measures of 61 MD patients, 64 migraine patients, and 55 control, we used 56, 59, and 42 measures respectively.

## 3. Results

### 3.1. Clinical Profile in Early and Late Onset MD

Table 2 includes all patients with EOMD. Twenty-three patients had been diagnosed as unilateral MD (54.5%) and 21 were considered bilateral MD (45.5%). Twenty-six (59%) were women and 18 (41%) were men.

Table 3 shows all individuals with LOMD. Twenty-six (67%) had been diagnosed as unilateral MD and 13 had bilateral MD (33.3%). Twenty-one (54%) were women and 18 (46%) men.

There was no difference in the frequency of FMD between patients with EOMD (11 patients, 25%) and LOMD (11 patients, 28%). Eighteen years was the mean duration of the disease in EOMD patients, whereas in LOMD patients the mean was 9.8 years old. In our series, elevated levels of IL-1β (>4.78 pg/mL) were observed in 5 of 44 (11.3%) patients with EOMD and in 4 of 39 (10.3%) patients with LOMD.

### 3.2. EOMD Has a Higher Prevalence of Migraine Than LOMD Patients

Table 4 compares the clinical features of patients with EOMD and LOMD. Patients with EOMD had a longer duration of the disease, defined as the time since the onset of the symptoms than patients with LOMD (*p* = 0.004). As expected, both groups of MD patients had similar hearing loss thresholds, familial history of hearing loss, and prevalence of vascular risk factors. However, EOMD patients presented a higher frequency of migraine than LOMD patients (*p* = 0.045), but no differences were found for any type of headache between both groups.

### 3.3. Cytokine Levels Do Not Allow to Distinguish between EOMD and LOMD Patients

Cytokines were measured in 56 MD patients (40 EOMD and 16 LOMD). We did not observe any differences between the levels of IL-1β or any of the chemokines CCL3, CCL4, CXCL1, CCL22, CCL8 (all *p*-values > 0.1, (Appendix A). However, when we performed a sex-specific analysis, there were significant differences in EOMD for IL-1β, with higher values in men (*p* = 0.039), and for CXCL1 with higher values in women (*p* = 0.007, Figure 1). Moreover, CCL3 levels were significantly higher in LOMD women than men (*p* = 0.04).

### 3.4. Cytokines Levels Do Not Differ between Patients with and without Migraine in MD

Since patients with EOMD presented a higher prevalence of migraine than LOMD (*p* = 0.045), we compared cytokines levels considering migraine as a clinical covariable. However, none of the cytokines or chemokines measured showed significant differences between patients with and without migraine in MD (Figure 2). Interestingly, the prevalence of a comorbid autoimmune disease was significantly higher in patients with MD and migraine compared to patients without migraine (*p* < 0.001).

We also separated MD patients according to IL-1β levels into two groups: MD with high levels of IL-1β (MDH) and low levels of IL-1β (MDL) (Figure 3). However, we found no difference in the relative frequency of patients according to the clinical subgroups and IL-1β levels (Table 5).

### 3.5. Cytokines Levels Do Not Differ between Familial (FMD) and Sporadic (SMD) MD Patients

We observed that the IL-1β levels were not different between familial and sporadic MD, however, significant differences were found in CXCL1 (*p* = 0.024) and CCL22 (*p* = 0.012), (Figure 4). Moreover, CXCL1 and CCL22 levels showed a higher correlation in FMD (*r* = 0.62) than in sporadic cases (*r* = 0.37).

### 3.6. CCL18, CCL22, and CCL4 Levels Are Different between Patients with MD, Migraine, and Controls

Levels of IL-1β, CCL3, CCL4, CXCL1, CCL22, and CCL8 were also measured in 55 healthy controls and 64 migraine patients. Significant differences were found in the levels of CCL18 (*p* = 0.001), CCL22 (*p* = 0.03), and CCL4 (*p* = 0.02) when we compared MD, migraine, and controls; however, no differences were found in IL-1β or CCL18 levels of MD vs. migraine patients (Figure 5).

We could observe that the levels of CCL18 were higher in patients with MD or migraine when they were compared to controls. However, CCL18 levels did not allow to distinguish between MD and migraine (*p* = 0.74). Moreover, no correlation was found in CCL18 levels and the age of the patients, either in controls (*p* = 0.59), patients with MD (*p* = 0.43), or migraine (*p* = 0.23) (Appendix A).

## 4. Discussion

This study analyzes the clinical profile and the levels of several cytokines and chemokines (IL-1β, CCL3, CCL4, CCL22, CCL18, and CXCL1) in patients with MD, according to the age of onset of the disease. The age of onset in MD follows a normal distribution [4,5] with a mean of ~43 years old. We selected Spanish MD patients < 35 and >50 years from the Meniere Disease Consortium to investigate clinical features in patients with early- and late-onset, respectively. The time of onset was confirmed from the medical records

The main finding is that migraine is more frequently observed in patients with EOMD than in LOMD, but the levels of cytokines are not related to the age of onset in MD.

Previous studies have reported that ~20% of patients with MD show high levels of pro-inflammatory cytokines, including IL-1β, TNF-α, and IL-6 [9]. Moreover, Flook et al. (2019) revealed significant differences in several cytokines and chemokines between patients with MD and vestibular migraine, suggesting that this set of protein biomarkers could distinguish MD and vestibular migraine [10].

The findings in this study do not support the hypothesis that the peripheral profile of immune molecules is related to the age of onset in MD or the presence of migraine. Therefore, we measured the levels of IL-1β, CCL3, CCL4, CXCL1, CCL22, and CCL18 in a cohort of 83 patients with MD, which included 44 EOMD and 39 LOMD, and 64 patients with migraine without vestibular symptoms to control the effect of migraine itself on cytokine levels.

We tested the hypothesis that activation of IL-1β signaling contributes to maintaining an autoinflammatory response in MD, however, this response is not age-dependent. Considering that we observed a cytokine increase in PBMCs extracted from peripheral blood, this could indicate that a subgroup of patients with MD has a systemic pro-inflammatory response, which could explain some patient’s responses to anti-inflammatory drugs [15,16]. Our study shows that IL-1β concentrations are not age-dependent, neither in MD patients nor healthy controls. Of note, we also found that IL-1β levels are higher in a subgroup of patients with migraine, suggesting a pro-inflammatory state in these patients.

We observed that EOMD patients have a higher prevalence of migraine as a comorbid condition compared to LOMD. This finding confirms the observation reported by Frejo et al. that patients with MD and migraine (type 4) presented an earlier onset than the rest of patients with MD [4,5].

In addition, the clinical association between autoimmunity and MD suggests the possibility of an autoimmune endophenotype according to the immune role of the endolymphatic sac or the synthesis of cytokines [17]. There are two different endotypes of MD patients, according to the pathology of the endolymphatic sac: one endotype characterized by the epithelial degeneration of the endolymphatic sac and a second endotype showing hypoplasia of the endolymphatic sac [18]; however, the cytokine profile or the association with migraine with the hypoplastic or degenerative endolymphatic sac has not been investigated.

To our knowledge, this is the first study in which inflammatory biomarkers and clinical phenotypes were simultaneously assessed both in adult men and women with early-onset as well as late-onset MD. We found an overall reduction in the systemic levels of inflammatory markers in men with EOMD or LOMD, where significant differences were observed for IL-1β, CXCL1, and CCL3. Moreover, we observed no significant differences in other inflammatory biomarkers between groups.

Familial MD has an earlier onset than sporadic cases, suggesting that early-onset patients might have a higher genetic burden, including patients with a potential recessive inheritance. Previous studies have shown that there is an enrichment of rare missense variants in some genes related to hearing loss, such as: *GJB2*, *USH1G*, *SLC26A*, *USH1G*, *ESRRB*, and *CLDN14*, as well as in some genes implicated in the “axonal guidance signaling” signaling pathway such as *NTN4* and *NOX3*, in patients with sporadic MD [19]. This excess of missense variants in some genes may increase the risk of developing hearing loss in MD [20]. We have found higher levels of CXCL1 and CCL22 in familial MD, but the relevance of this finding is unclear. The chemokine CXCL1 recruits and activates neutrophils to the site of infection [21], and it is essential for the expression of pro-inflammatory mediators, activation of NF-kB, and MAPKs [22]. CCL22 is a macrophage-derived chemokine and a potent chemoattractant for chronically activated T-lymphocytes [23,24]. CCL22 is thought to be involved in several conditions, including allergy, autoimmunity, and tumor growth [24], and increased CCL22 expression has been reported in autoinflammatory skin diseases [25].

There are several mechanisms that may drive the onset of MD. A persistent inflammatory response may affect one or both ears and local factors could be involved. The accumulation of endolymph is the result of inner ear damage usually associated with sensorineural hearing loss [26].

The autoinflammatory disease of the inner ear might include an innate immune response mediated by monocytes, IL-1β, and non-specific autoantibodies [27]. In healthy individuals, IL-1 RA is detectable in plasma and IL-1β levels are usually not detectable. IL-1 RA levels were more elevated in most patients with MD regardless of the levels of IL-1β [9]. It was also observed that only 58 genes were differentially expressed in MD patients with high basal levels of IL-1β, supporting the hypothesis that this abnormal response may facilitate the identification of a subgroup of MD patients with a specific gene expression profile [9].

Cytokines are considered as pain mediators in neurovascular inflammation, and therefore, they may cause the generation of migraine pain [28]. Changes in inflammatory activity in migraine patients may be one of the main causes of endothelial dysfunction. The increase in endothelial cell permeability leads to an increase in the paracellular leakage of plasma fluid and proteins [29].

Different findings have been reported in the clinical studies measuring cytokine levels in migraine patients [30,31,32,33]. These studies have indicated that immunological changes could play a role in the pathophysiology of migraine. We can speculate that increased release of CCL18 observed in patients with migraine in combination with other pro-inflammatory mediators is likely to contribute to a persistent inflammatory state that may trigger the onset of MD in susceptible individuals. This subgroup of patients with MD and migraine may have an earlier onset and higher risk of autoinflammation.

In addition, the lack of correlation between age and IL-1β plasma levels, regardless of the important role of aging in the immune response, suggests that genetic or epigenetic factors are needed to develop MD in patients with migraine [34].

Our study shows that a better understanding of how the immune mediators could influence the age of onset in MD disease is of great relevance to provide clues on potential strategies to delay the onset of MD in susceptible individuals in an increasingly aged population.

## 5. Limitations of the Study

The main limitation of our study is that the number of investigated patients was low (*n* = 83), and further patients will be needed to validate our findings. Our study is limited by the cross-sectional design and the small number of cytokines and clinical data collected, so it may be not enough to classify patients with MD according to the age of onset. Moreover, the lack of correlation between the age of individuals and IL-1β or CCL18 levels suggest inter-individual differences associated with multiple factors, including migraine.

## 6. Conclusions

Patients with EOMD have a higher prevalence of migraine than patients with LOMD.The levels of IL-1β or chemokines were not dependent on the age of the individuals or the presence of migraine in patients with MD.The levels of CCL18, CCL22, and CCL4 were different between patients with MD or migraine and controls.

## Figures and Tables

**Figure 1 jcm-10-04052-f001:**
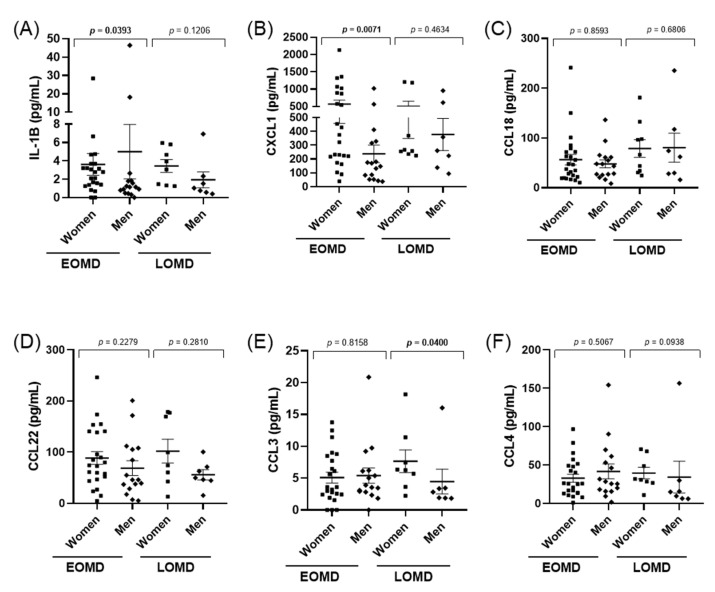
Scatter plots showing medians and IQRs for IL-1β (**A**), CXCL1(**B**), CCL18 (**C**), CCL22 (**D**), CCL3 (**E**), CCL4 (**F**), levels measured in PBMC’s supernatant in patients with EOMD and LOMD. Cytokine and chemokine levels were compared between men and women by the Mann–Whitney test and *p*-values are shown for each comparison. Significant differences (*p* < 0.05) are highlighted in bold.

**Figure 2 jcm-10-04052-f002:**
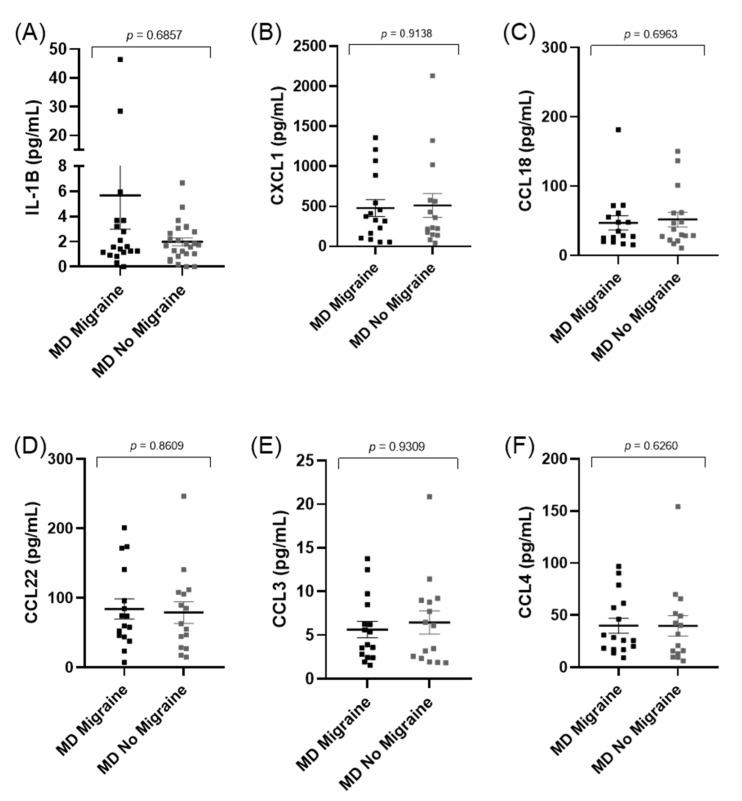
Scatter plots showing cytokine and chemokines levels in patients with MD (**A**–**F**), stratified according to the presence of migraine.

**Figure 3 jcm-10-04052-f003:**
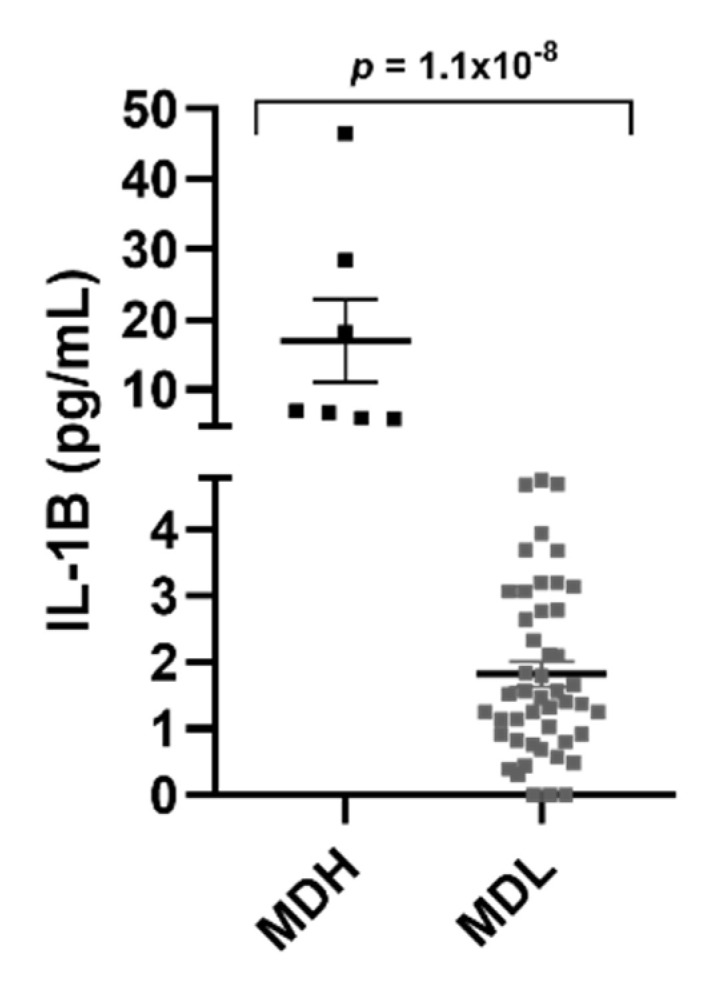
MD patients separated according to their IL-1β levels (MDH (>4.78 pg/mL) and MDL (<4.78 pg/mL)).

**Figure 4 jcm-10-04052-f004:**
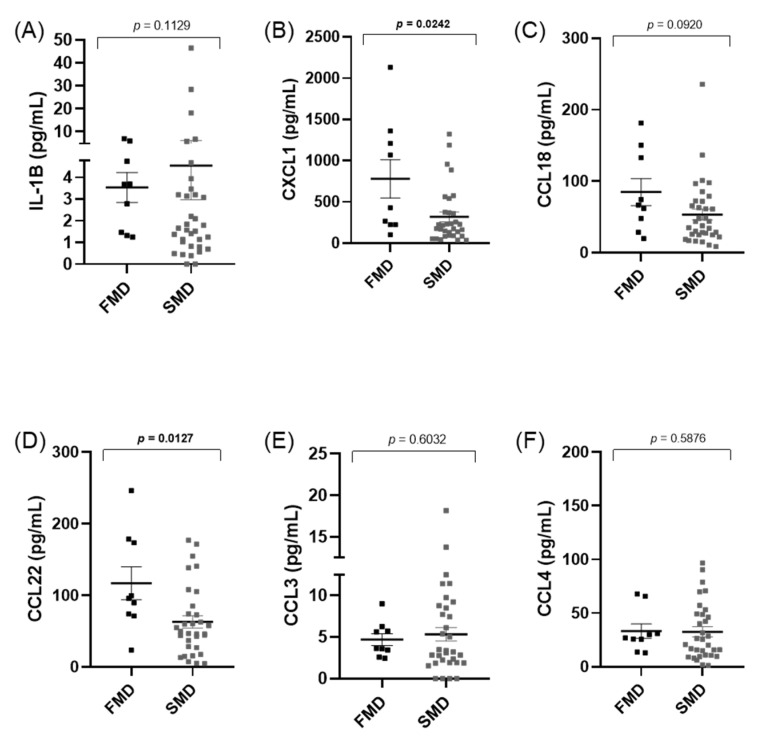
Cytokines and chemokine levels in patients with familial or sporadic MD (**A**–**F**). FMD: Familial Meniere disease Significant differences; SMD: Sporadic Meniere disease; (*p* < 0.05) are highlighted in bold.

**Figure 5 jcm-10-04052-f005:**
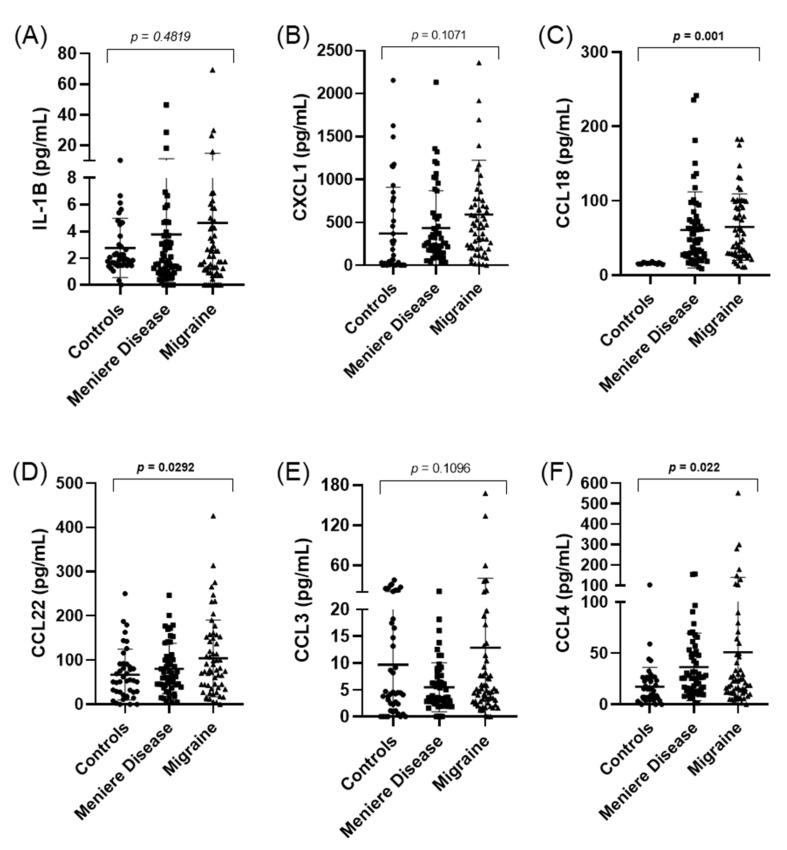
Scatter plots showing cytokines with the concentrations of IL-1β (**A**), CXCL1 (**B**), CCL18 (**C**), CCL22 (**D**), CCL3 (**E**), and CCL4 (**F**) in healthy controls, patients with MD, and migraine. Significant differences (*p* < 0.05) are highlighted in bold.

**Table 1 jcm-10-04052-t001:** Clinical subgroups of patients with unilateral and bilateral Meniere disease. Modified from Frejo et al. [4,5].

Unilateral Ménière’s Disease (UMD)
Type 1: Sporadic and classical MD.
Type 2: Delayed MD (hearing loss precedes vertigo attacks in months or years).
Type 3: Familial MD (at least two individuals with MD related in the first or second degree).
Type 4: Sporadic MD with migraine (temporal relationship not required).
Type 5: Sporadic MD plus an autoimmune disease.
**Bilateral Ménière’s Disease (BMD)**
Type 1: Unilateral hearing loss becomes bilateral.
Type 2: Sporadic, simultaneous hearing loss (usually symmetric).
Type 3: Familial MD (most families have bilateral hearing loss, but unilateral and bilateral cases may coexist in the same family).
Type 4: Sporadic MD with migraine.
Type 5: Sporadic MD with an autoimmune disease.

**Table 2 jcm-10-04052-t002:** Clinical features and basal levels of IL-1β cytokine in patients with EOMD (*n* = 44). Elevated levels of IL-1β are highlighted in bold type.

Patient	Gender	Ear	Age of Onset	Duration of Disease (y)	FMD	Migraine	MD Subgroup	IL-1β Level
1	Woman	Bilateral	34	8	Yes	Yes	3	3.68
2	Woman	Bilateral	33	8	Yes	Yes	3	1.24
3	Woman	Unilateral	28	12	No	Yes	4	0.82
4	Man	Unilateral	29	9	No	No	1	1.57
5	Woman	Bilateral	33	24	No	Yes	4	3.20
6	Woman	Bilateral	18	34	Yes	No	3	4.74
7	Woman	Unilateral	28	27	No	X	4	1.41
8	Woman	Unilateral	20	1	Yes	Yes	3	**18.64**
9	Woman	Unilateral	28	8	No	No	2	3.69
10	Woman	Unilateral	29	20	Yes	Yes	3	2.79
11	Man	Bilateral	27	16	No	No	2	2.64
12	Woman	Bilateral	33	8	No	Yes	4	2.10
13	Woman	Bilateral	24	15	No	Yes	4	**28.45**
14	Woman	Bilateral	33	17	No	No	5	1.66
15	Woman	Bilateral	23	18	No	Yes	4	1.59
16	Man	Bilateral	31	15	No	Yes	1	1.25
17	Man	Unilateral	34	8	No	No	1	0.49
18	Woman	Unilateral	27	17	No	No	1	2.11
19	Man	Bilateral	35	16	No	No	1	0.92
20	Woman	Bilateral	34	20	No	No	1	3.68
21	Woman	Unilateral	35	20	Yes	No	3	0.01
22	Man	Bilateral	29	15	No	No	2	0.18
23	Man	Unilateral	29	11	No	No	1	0
24	Man	Unilateral	24	1	No	No	1	1.84
25	Man	Unilateral	30	27	No	Yes	1	**46.41**
26	Man	Bilateral	27	22	No	No	1	0
27	Man	Unilateral	32	3	No	No	1	2.77
28	Man	Bilateral	22	23	Yes	Yes	1	0.44
29	Woman	Bilateral	31	5	No	Yes	1	1.14
30	Man	Bilateral	22	22	No	No	1	1.57
31	Man	Bilateral	25	25	No	No	1	0.31
32	Man	Bilateral	34	34	No	No	1	1.14
33	Man	Bilateral	28	18	No	No	1	**18.18**
34	Woman	Unilateral	28	25	No	No	5	2.19
35	Woman	Unilateral	23	50	No	No	1	3.14
36	Woman	Unilateral	29	30	No	No	5	**6.66**
37	Woman	Unilateral	23	50	No	No	1	3.07
38	Man	Unilateral	27	5	No	No	1	1.79
39	Woman	Unilateral	18	18	Yes	Yes	3	4.68
40	Woman	Unilateral	35	20	No	No	1	0
41	Woman	Unilateral	21	2	Yes	No	3	0
42	Man	Unilateral	24	28	Yes	No	3	0.80
43	Woman	Unilateral	18	9	Yes	Yes	2	0.69
44	Woman	Unilateral	24	28	No	Yes	4	1.37

**Table 3 jcm-10-04052-t003:** Clinical features and basal levels of IL-1β cytokine in patients with LOMD (*n* = 39). Elevated levels of IL-1β are highlighted in bold type.

Patient	Gender	Ear	Age of Onset	Duration of Disease (y)	FMD	Migraine	MD Subgroup	IL-1β Level
1	Man	Bilateral	55	20	No	No	2	0.83
2	Woman	Unilateral	65	8	Yes	No	3	1.23
3	Man	Unilateral	50	13	No	No	1	1.56
4	Man	Bilateral	55	30	No	No	2	2.24
5	Man	Unilateral	61	16	No	No	1	2.64
6	Woman	Unilateral	50	23	Yes	No	3	2.10
7	Man	Bilateral	53	16	No	No	2	3.14
8	Woman	Unilateral	50	8	No	No	1	2.67
9	Man	Bilateral	53	30	No	No	2	4.34
10	Man	Unilateral	50	14	Yes	No	3	**6.93**
11	Man	Unilateral	62	1	No	Yes	4	0.01
12	Woman	Unilateral	52	17	No	Yes	4	0.57
13	Woman	Bilateral	56	24	No	No	2	2.27
14	Woman	Unilateral	55	10	No	No	1	2.06
15	Woman	Unilateral	63	13	Yes	Yes	3	1.07
16	Man	Bilateral	56	5	No	No	2	0.58
17	Man	Unilateral	53	2	Yes	Yes	3	0.01
18	Man	Bilateral	54	3	No	No	2	1.03
19	Woman	Bilateral	53	4	No	Yes	4	2.67
20	Man	Bilateral	55	17	Yes	No	3	1.27
21	Woman	Unilateral	50	20	Yes	No	3	1.73
22	Woman	Bilateral	55	9	Yes	Yes	3	**5.95**
23	Woman	Unilateral	50	8	No	No	1	3.07
24	Woman	Bilateral	52	6	No	No	2	1.03
25	Man	Bilateral	53	2	Yes	Yes	3	3.68
26	Woman	Unilateral	56	2	No	No	1	0.83
27	Man	Unilateral	57	2	No	No	1	1.66
28	Man	Bilateral	53	7	No	No	2	0.39
29	Woman	Unilateral	65	1	No	Yes	4	**4.96**
30	Man	Unilateral	69	2	No	No	1	0.76
31	Man	Unilateral	66	5	No	No	1	2.33
32	Woman	Unilateral	51	5	No	No	1	4.67
33	Woman	Unilateral	63	6	No	No	1	3.94
34	Woman	Unilateral	61	6	No	No	1	1.25
35	Man	Unilateral	51	7	No	No	1	1.52
36	Woman	Unilateral	50	8	No	No	1	**5.79**
37	Woman	Unilateral	54	1	Yes	No	3	1.32
38	Woman	Unilateral	57	6	Yes	No	3	1.47
39	Woman	Unilateral	54	7	No	Yes	1	3.47

**Table 4 jcm-10-04052-t004:** Clinical features of patients with EOMD and LOMD included in the study. Migraine was more frequently observed in patients with EOMD.

Variable	EOMD (*n* = 44)	LOMD (*n* = 39)	*p*-Value
Duration of the disease (mean ± SD)	18.0 ± 11.4	9.8 ± 7.9	**0.0004**
Age of onset (mean ± SD)	27.7 ± 4.9	55.6± 5.2	**<0.0001**
Sex (% female)	59.0	53.8	0.195
Bilateral hearing loss (%)	45.5	33.3	0.545
Hearing loss (% synchronic)	29.5	12.8	0.338
MD type			0.589
1	21	15	
2	4	9	
3	9	4	
4	7	3	
5	3	0	
Hearing loss (%)			0.357
High frequency	61.4	69.2	
Low frequency	38.6	30.8	
Family History of hearing loss (%)	25.0	28.2	0.999
Any Headache (%)	59.0	56.4	0.440
Migraine (%)	**38.6**	**23.0**	**0.045**
Arterial Hypertension (%)	22.7	25.6	0.948
Smoking (%)	13.6	10.3	0.532
Dyslipidemia (%)	18.2	28.2	0.400
Diabetes Mellitus (%)	25.0	25.6	0.854

Significant differences (*p* < 0.05) are highlighted in bold.

**Table 5 jcm-10-04052-t005:** Frequency distribution of patients with MD according to the clinical subgroup and IL-1β levels.

Variable	MDH (*n* = 9)	MDL (*n* = 74)	*p*-Value
MD type (%)			0.9924
1	3 (33.3)	33 (44)	
2	0 (0)	13 (14.6)	
3	3 (33.3)	17 (22.9)	
4	2 (22.2)	9 (12.1)	
5	1 (11.1)	2 (2.7)	

## Data Availability

All data generated or analysed during this study are included in this article and its Appendix A.

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
