# Peer review of "Clinical and Cytokine Profile in Patients with Early and Late Onset Meniere Disease"

_jcm, 2021, doi:10.3390/jcm10184052_

Round 1

Reviewer 1 Report

This is a very interesting well designed and well conducted study dealing with the possibility to check if some inflammatory markers could explain the different age of onset in MD patients, both in unilateral and bilateral inner ear involvement.

The methods and results sections are well organized and excellently explained with the help of explanatory images.

Only an observation: the discussion seems to me not completely focused on the aim of the study (characterize the clinical and cytokine profile, according to the age of onset in MD). Some considerations seem to be too speculative (for example the role of the endolymphatic sac or the role of the neuroinflammation): for the reader is useful that the Authors point out on the absence of significant differences of Cytokines and chemokine levels in the two groups of MD patients (early and late onset) and in patients with VM

Author Response

Reviewer 1

This is a very interesting well designed and well conducted study dealing with the possibility to check if some inflammatory markers could explain the different age of onset in MD patients, both in unilateral and bilateral inner ear involvement.

The methods and results sections are well organized and excellently explained with the help of explanatory images.

Thank you for your kind comments.

Only an observation: the discussion seems to me not completely focused on the aim of the study (characterize the clinical and cytokine profile, according to the age of onset in MD). Some considerations seem to be too speculative (for example the role of the endolymphatic sac or the role of the neuroinflammation): for the reader is useful that the Authors point out on the absence of significant differences of Cytokines and chemokine levels in the two groups of MD patients (early and late onset) and in patients with VM.

We have re-written the discussion section to remark this point in the second paragraph (page 11, line 254): “The main finding is that migraine is more frequently observed in patients with EOMD than in LOMD, but the levels of cytokines are not related with the age of onset in MD”. Moreover, this is also stated in the conclusion section: The levels of IL-1β or chemokines were not dependent on the age of the individuals or the presence of migraine in patients with MD.

Reviewer 2 Report

Authors investigated clinical and cytokine differences between patients with early and late-onset of Meniere's disease (MD) and they also evaluated the association with the MD subgroup proposed by the author group. However, the results do not support their hypothesis that early-onset MD cases and late-onset MD cases have different cytokine profiles that reflect the different pathogenesis of both groups.  Also, it is understandable that some MD cases may have autoimmune or autoinflammatory pathologies. However, by nature, the pathogenetic cause of MD is unknown and may be diverse. Therefore, I think that the clinical interest is whether the prognosis of cochlear and vestibular functions is poor in some cases with a background of autoimmunity or autoinflammation pathology.  Some points I noticed are described following.

Major comments

Authors selected migraine cases without vestibular symptoms, but how many patients have been diagnosed with migraine with vestibular symptoms, especially so-called vestibular migraine? The difference in the pathological condition between MD and vestibular migraine may be important in clinical practice.

They separated into two groups based on the onset age of MD, but I do not understand why they defined less than 35 years old as the early onset and more than 50 years old as late onset.  They should explain the evidence or basis.  Also, how did they confirm the time of onset from the medical records or healthcare file, or self-report?

Why didn’t they measure cytokines for all subjects, especially in this study the number of patients in the late-onset group who were measured the cytokines were small.  They should measure the cytokine level in more patients with late-onset MD.

Familiar MD showed high levels in CXCL1 and CCL22. What does this mean? Also, were these correlated with CCL18 and CCL4? 

Are most subjects in this paper the same as the population of patients in the previously published papers of this group (e.g. Frejo et al.) ? 

They described that persistent inflammation is involved in the onset of MD. What is the mechanism? If it is true, why most MD cases show unilateral, even in a natural course?  How do they explain an association with endolymphatic hydrops formation which is the most popular pathology of MD?

The hypothesis that autoimmune inflammatory disease and autoimmune disease are involved in systemic and bilateral of MD is understandable, but authors demonstrated that early-onset cases have higher prevalence of migraine, and migraine patients showed higher cytokines, I wonder why no difference in cytokine level and no tendency to be bilateral MD between both groups were found.  I question if the onset time of MD has a clinical significance or not.

I have some questions, foe example,  the hypothesis and the conclusion are made from the data collected in the previous study, and they simply compared the date of two groups that were divided by the median onset age. In clinical practice, most MD cases have good prognosis regardless of whether they develop early or late.  Bilaterality and difficulty to control hearing prognosis and vertigo attacks are most critical issue for MD patients. I question how the different cytokine profiles affect the clinical course of MD.

Minor comments

Line66:  they should correct “than” into “that”.

Line 100: authors described that THI was collected in the method part, but I could not find the results.

Was there a difference in the average age between early-onset and late-onset groups?

Line 179:  What kind of autoimmune comorbidity did early-onset and late-onset MD groups have? Of them,  they should describe how many cases have high IL-b or how many patients have bilateral MD. 

Author Response

Reviewer 2

Authors investigated clinical and cytokine differences between patients with early and late-onset of Meniere's disease (MD) and they also evaluated the association with the MD subgroup proposed by the author group. However, the results do not support their hypothesis that early-onset MD cases and late-onset MD cases have different cytokine profiles that reflect the different pathogenesis of both groups.  Also, it is understandable that some MD cases may have autoimmune or autoinflammatory pathologies. However, by nature, the pathogenetic cause of MD is unknown and may be diverse. Therefore, I think that the clinical interest is whether the prognosis of cochlear and vestibular functions is poor in some cases with a background of autoimmunity or autoinflammation pathology.  Some points I noticed are described following.

Thank you for your detailed review. We fully agree with the reviewer and the main conclusion is that we have not found any difference in the cytokine or chemokine profile between early and late onset MD. We also agree that the clinical interest is if autoinflammation or autoimmunity are associated with a poor prognosis in hearing or vestibular function. Our study is a cross-sectional study and we will need a prospective longitudinal study to assess this hypothesis.

Major comments

Authors selected migraine cases without vestibular symptoms, but how many patients have been diagnosed with migraine with vestibular symptoms, especially so-called vestibular migraine? The difference in the pathological condition between MD and vestibular migraine may be important in clinical practice.

In this study we did not investigate any patients with vestibular migraine (VM). We compared cytokine profile between patients with MD and VM a previous study and found significant differences in cytokine and chemokine profile (Flook et al, 2019)

They separated into two groups based on the onset age of MD, but I do not understand why they defined less than 35 years old as the early onset and more than 50 years old as late onset.  They should explain the evidence or basis.  Also, how did they confirm the time of onset from the medical records or healthcare file, or self-report?

The age of onset in MD follows a normal distribution (Frejo et al. 2016; 2017) with the mean around 43 years old. We selected Spanish MD patients < 35 and > 50 years from the Meniere disease Consortium to investigate clinical features in patients with early and late onset respectively. The time of onset was confirmed from the medical records. We state this on page 11, lines 250-53.

Why didn’t they measure cytokines for all subjects, especially in this study the number of patients in the late-onset group who were measured the cytokines were small.  They should measure the cytokine level in more patients with late-onset MD.

We could measure cytokines in 56 patients with MD (40 EOMD and 16 LOMD). We agree that the number of samples from patients with LOMD is small, but no significant difference was observed (Figure A1). We have stated this in the limitations of the study (page 13, line 329).

Familiar MD showed high levels in CXCL1 and CCL22. What does this mean? Also, were these correlated with CCL18 and CCL4?

As we have stated in the discussion “We have found in higher levels of CXCL1 and CCL22 in familial MD, but the relevance of this finding is unclear. The chemokine CXCL1 plays recruiting and activating neutrophils to the site of infection [21], and it is essential for the expression of proinflammatory mediators, activation of NF-kB and MAPKs [22]. CCL22 is a macrophage-derived chemokine and a potent chemoattractant for chronically activated T-lymphocytes [23, 24]. CCL22 is thought to be involved in several conditions, including allergy, autoimmunity and tumor growth [24] and increased CCL22 expression has been reported in autoinflammatory skin diseases [25]. 

“Moreover, CXCL1 and CCL22 levels showed a higher correlation in FMD (r=0.62) than in sporadic cases (r=0.37).” This sentence was added on page 9, line 227. CXCL1 and CCL22 were not correlated with CCL18 or CCL4.

Are most subjects in this paper the same as the population of patients in the previously published papers of this group (e.g. Frejo et al.) ? 

We included 83 patients with MD in this study. Eighteen of them were also included in the study performed by Flook et al. (2019). We have added this information in the methods on page 3, line 88.

They described that persistent inflammation is involved in the onset of MD. What is the mechanism? If it is true, why most MD cases show unilateral, even in a natural course?  How do they explain an association with endolymphatic hydrops formation which is the most popular pathology of MD?

There are several mechanisms that may drive the onset of MD. A persistent inflammatory response may affect one or both ears and local factors could be involved. The accumulation of endolymph is the result of an inner ear damage usually associated with sensorineural hearing loss (Tonndorf, 1976). We have added this paragraph in the discussion section on page 12, lines 303-306.

The hypothesis that autoimmune inflammatory disease and autoimmune disease are involved in systemic and bilateral of MD is understandable, but authors demonstrated that early-onset cases have higher prevalence of migraine, and migraine patients showed higher cytokines, I wonder why no difference in cytokine level and no tendency to be bilateral MD between both groups were found.  I question if the onset time of MD has a clinical significance or not.

Figure 2 shows that cytokine and chemokine levels were not different in patients with MD with or without migraine. Figure 5 shows that cytokine levels were not different between MD and migraine and CCL18 levels were higher in both MD and migraine patients when compared to controls. Figure A2 also shows that age does not influence CCL18 levels neither MD nor migraine patients. The main conclusion is that EOMD is associated with a high frequency of migraine than LOMD, but cytokine levels do not explain this association.

I have some questions, for example,  the hypothesis and the conclusion are made from the data collected in the previous study, and they simply compared the date of two groups that were divided by the median onset age. In clinical practice, most MD cases have good prognosis regardless of whether they develop early or late.  Bilaterality and difficulty to control hearing prognosis and vertigo attacks are most critical issue for MD patients. I question how the different cytokine profiles affect the clinical course of MD.

As we have stated above this is a cross-sectional study and we will need a prospective longitudinal study to assess if cytokine profile may predict the clinical course in MD.

Minor comments

Line66:  they should correct “than” into “that”.

We have corrected it.

Line 100: authors described that THI was collected in the method part, but I could not find the results.

This was a mistake, and we have corrected it. 

Was there a difference in the average age between early-onset and late-onset groups?

This information (age of onset and duration of the disease) was included in Table 4.

Line 179:  What kind of autoimmune comorbidity did early-onset and late-onset MD groups have? Of them,  they should describe how many cases have high IL-b or how many patients have bilateral MD. 

There were 3 patients with autoimmune comorbidities, all were classified as EOMD (Patients 14, 34 and 36 in Table 2). Patient 14 presented bilateral MD and anti-phospholipid syndrome. Patients 34 and 36 were unilateral MD and had autoimmune hypothyroidism.

Round 2

Reviewer 2 Report

Authors have responded to all commnets, and the paper has been improved.

I have no more comment.

Thank you for your big job.